# Magnetite Microspheres for the Controlled Release of Rosmarinic Acid

**DOI:** 10.3390/pharmaceutics14112292

**Published:** 2022-10-26

**Authors:** Cristina Chircov, Diana-Cristina Pîrvulescu, Alexandra Cătălina Bîrcă, Ecaterina Andronescu, Alexandru Mihai Grumezescu

**Affiliations:** 1Department of Science and Engineering of Oxide Materials and Nanomaterials, University Politehnica of Bucharest, 011061 Bucharest, Romania; 2National Research Center for Micro and Nanomaterials, University Politehnica of Bucharest, 060042 Bucharest, Romania; 3Faculty of Medical Engineering, University Politehnica of Bucharest, 060042 Bucharest, Romania; 4Academy of Romanian Scientists, 54 Spl. Independentei, 050045 Bucharest, Romania; 5Research Institute of the University of Bucharest—ICUB, University of Bucharest, 050657 Bucharest, Romania

**Keywords:** magnetite, rosmarinic acid, anticancer, natural biocompounds, polyphenols, polyethylene glycol

## Abstract

Since cancer incidence is constantly increasing, novel and more efficient treatment methods that overcome the current limitations of chemotherapy are continuously explored. In this context, the aim of the present study was to investigate the potential of two types of magnetite microspheres as drug delivery vehicles for the controlled release of rosmarinic acid (RA) in anticancer therapies. The magnetite microspheres were obtained through the solvothermal method by using polyethylene glycol (PEG) with two different molecular weights as the surfactant. The physicochemical characterization of the so-obtained drug delivery carriers involved X-ray diffraction (XRD) coupled with Rietveld refinement, scanning electron microscopy (SEM), Fourier transform infrared spectroscopy (FT-IR), dynamic light scattering (DLS) and zeta potential, and UV–Vis spectrophotometry. The magnetite-based anticancer agents were biologically evaluated through the ROS-Glo H_2_O_2_ and MTT assays. Results proved the formation of magnetite spheres with submicronic sizes and the effective RA loading and controlled release, while the biological assays demonstrated the anticancer potential of the present systems. Thus, this study successfully developed a promising drug delivery alternative based on magnetite that could be used in the continuous fight against cancer.

## 1. Introduction

With a constantly increasing incidence, cancer represents one of the major causes of concern for the global health system, with more than 19 million cases reported in 2020 alone. It is also the most common cause of death, with the predictions of the World Health Organization estimating around 13 million annual deaths by 2030 [1,2,3]. Although there are many treatment methods available to increase the survival rate, such as chemotherapy, radiotherapy, immunotherapy, or surgical interventions, they are characterized by various limitations, e.g., non-specific targeting, high and frequent dosages, occurrence of drug resistance, and poor distribution [2,4].

Generally, traditional drug formulations ensure the release of drugs through diffusion or drug solubility, but not the protection of the drug against environmental degradation or premature release [5,6,7]. Therefore, a significant amount of research has been directed towards the development of drug delivery alternatives that could overcome such limitations. Drug delivery systems represent engineered devices applied for the targeted transport of bioactive compounds at the sites of disease where they release the therapeutic cargo in a controlled manner [8,9]. In this context, micro/nanoparticles have emerged as a promising platform for the delivery of anticancer agents that ensure the control of the rate, time and site of action and, thus, facilitate patient compliance [2,4,10,11,12]. Furthermore, the versatility of particle-based carriers allows the tuning of their features, such as size, shape, solubility, stability, degradability, and surface reactivity, which in turn direct the pharmacokinetics and pharmacodynamics of the administered biocompounds [4].

Iron oxide represents a family of materials that have received tremendous attention within the biomedical area, as their safety resides on the iron component, which is an essential element of the human organism [5]. Their biomedical applicability is quite varied, since they have been investigated as targeted drug delivery micro- and nano-vehicles, hyperthermia agents, diagnostic and theranostic platforms, or instruments for in vitro analysis techniques [13,14,15,16]. The most intensively studied and biocompatible form of iron oxide is magnetite (Fe_3_O_4_), which is also characterized by paramagnetic properties [5]. Moreover, its large surface-to-volume ratio allows for the adsorption or immobilization of large quantities of drug molecules, and its properties can be easily modified by applying different synthesis methods or parameters [17,18,19]. Nonetheless, in order to prevent agglomeration and to increase stability, water dispersibility, and cytocompatibility, magnetite micro/nanoparticles generally require surface modifications with numerous organic and inorganic structures [20]. In the context of cancer therapy, an NCBI database search using the combined keywords “magnetite” and “cancer” will retrieve a total of 1390 publications and 7 clinical trials published within the past 5 years [17]. Thus, the wide applicability of magnetite as vehicles for the delivery of anticancer agents has been demonstrated.

From another point of view, research is also focusing on exploring and developing novel bioactive molecules with anticancer activity, with current trends shifting towards the application of natural, plant-based biocompounds. Rosmarinic acid (RA) is a natural polyphenol, i.e., an ester of caffeic acid and 3-(3,4-dihydroxyphenyl)lactic acid, extracted from numerous plants of the *Boraginaceae* and *Lamiaceae* families that has been intensively investigated within the biomedical area [21,22,23,24,25,26]. RA is well-known for its biological effects when administered to the organism, namely antibacterial, antiviral, antimutagenic, and anti-inflammatory activities [27]. Additionally, there are numerous research studies demonstrating the anticarcinogenic potential of RA against a variety of cancer types [28,29,30,31,32,33,34].

Therefore, the main aim of the present study was to investigate the potential of two types of magnetite microsphere-based drug delivery carriers obtained through the solvothermal method for the controlled release of rosmarinic acid (RA) in anticancer therapy. The novelty of the study resides in the synthesis method which utilizes only one of the two iron precursors necessary for obtaining magnetite, as well as the possibility to obtain magnetite particles with submicronic sizes which are generally more stable and more difficult to develop. The design of the study including the synthesis and characterization methods are summarized in Figure 1.

## 2. Materials and Methods

### 2.1. Materials

Ferric chloride hexahydrate (FeCl_3_·6H_2_O), ethylene glycol anhydrous (C_2_H_6_O_2_), polyethylene glycol (PEG—C_2n_H_4n+2_O_n+1_) with two different molecular weights (Mw = 8000 g/mol and Mw = 20,000 g/mol), ethanol (C_2_H_6_O), and RA (C_18_H_16_O_8_) were purchased from Sigma-Aldrich Merck (Darmstadt, Germany). Sodium acetate trihydrate (NaAc—C_2_H_3_NaO_2_·3H_2_O) was purchased from Silal Trading (Bucharest, Romania). Phosphate buffer saline (PBS) was purchased from Carl Roth (Karlsruhe, Baden-Württemberg, Germany). All chemicals were of analytical purity and used with no further purification.

The BHK and HepG2 cell lines were used from the collection of the Institute for Diagnosis and Animal Health (I.D.A.H.)

### 2.2. Synthesis of Magnetite Microspheres and RA Loading

Magnetite microspheres were synthesized using the solvothermal reduction method, following the methodology described by Deng et al. [35]. Briefly, 6 g of FeCl_3_·6H_2_O was dissolved in 180 mL ethylene glycol, followed by the addition of 25.8 g NaAc and 4.5 g PEG. Two mixtures corresponding to each type of PEG used were obtained and magnetically stirred for 30 min. Subsequently, the mixtures were transferred into hermetically sealed polytetrafluoroethylene (Teflon)-lined stainless-steel autoclaves of 250 mL capacity. The autoclaves were heated and maintained at 200 °C for 8 h and allowed to cool at room temperature. The obtained black products were washed with ethanol three times through centrifugation at 6000 rpm for 10 min and dried at 40 °C for 12 h.

The microspheres were redispersed in 30 mL of ethanol, followed by the addition of RA in three different concentrations (1, 5, and 10%), and stirred for 24 h at 37 °C. Subsequently, the microspheres were centrifuged 6000 rpm for 10 min and dried at 40 °C for 12 h. The supernatant was kept for further analyses.

Table 1 summarizes the samples obtained and their coding for future references.

### 2.3. Physicochemical Characterization

#### 2.3.1. X-ray Diffraction (XRD) Coupled with Rietveld Refinement

The XRD analysis was performed on a PANalytical Empyrean diffractometer (PANalytical, Almelo, The Netherlands) equipped with a CuKα radiation. The samples were scanned within the 10–80° 2θ angle range, with a 0.5° incidence angle, a 0.0256° step size, and 255 s time per step. The Rietveld refinement algorithm for determining the crystallite size and unit cell parameters was conducted using the HighScore Plus software (version 3.0, PANalytical, Almelo, The Netherlands).

#### 2.3.2. Scanning Electron Microscopy (SEM)

The size and morphology of the magnetite microspheres were investigated through SEM, by placing a small amount of the powder onto a carbon band that was placed inside the analysis chamber of the Inspect F50 high-resolution microscope (Thermo Fisher—former FEI, Eindhoven, The Netherlands). The micrographs at different magnifications were acquired using secondary electrons with 30 KeV energy.

#### 2.3.3. Fourier Transform Infrared Spectroscopy (FT-IR)

The IR spectra were acquired using a Thermo Scientific Nicolet iS50 (Thermo Fischer Scientific, Waltham, MA, USA) spectrometer with a liquid nitrogen-cooled mercury cadmium telluride detector. The measurement was performed at room temperature using the attenuated total reflectance (ATR) mode, in the range of 4000–400 cm^−1^ and at a resolution of 4 cm^−1^. For each sample, 64 scans were co-added and processed using the OmnicPicta software (version 8.2, Thermo Nicolet, Thermo Fischer Scientific, Waltham, MA, USA).

#### 2.3.4. Dynamic Light Scattering (DLS) and Zeta Potential

The samples were homogenously dispersed in deionized water (~6.9 pH) at a concentration of 0.3 mg/mL using the ultrasonic bath for 10 min. A small amount was taken and placed inside the measurement cell of the DelsaMax Pro equipment (Backman Coulter, Brea, CA, USA). For each sample, measurements were performed in triplicate.

#### 2.3.5. UV–Vis Spectrophotometry

UV–Vis spectrophotometry was used both for determining the loading efficiency of RA within the magnetite microspheres and for estimating the RA release. A Thermo Evolution 600 double-beam UV–Vis spectrophotometer (Thermo Fischer Scientific, Waltham, MA, USA) was utilized for both types of measurements at a fixed wavelength (λ = 221 nm), at room temperature, using a standard glass cuvette with an optical path of 1 cm.

Initially, the quantity of the unloaded RA within the supernatant was determined and further used for calculating the drug loading capacity (LC%) and drug encapsulation efficiency (EE%) through the following equations:(1)$$\mathrm{RA}\;\mathrm{LC}\%=\frac{\mathrm{amount}\;\mathrm{of}\;\mathrm{entrapped}\;\mathrm{RA}}{\mathrm{total}\;\mathrm{amount}\;\mathrm{of}\;\mathrm{magnetite}\;\mathrm{microspheres}}\;\times \;100$$
(2)$$\mathrm{RA}\;\mathrm{EE}\%=\frac{\mathrm{total}\;\mathrm{amount}\;\mathrm{of}\;\mathrm{RA}\;-\;\mathrm{free}\;\mathrm{amount}\;\mathrm{of}\;\mathrm{RA}}{\mathrm{total}\;\mathrm{amount}\;\mathrm{of}\;\mathrm{RA}}\;\times \;100$$

Subsequently, for the RA release study, 100 mg of each sample was introduced into a dialysis bag and placed into a 25 mL PBS solution (pH = 7.4), which was maintained at 37 °C. At different timepoints (i.e., 1, 5, 10, 15, 30, and 60 min, 2, 3, 4, 8, 24, 48, and 72 h, and 7, 14, and 21 days), 2 mL of the supernatant was collected and replaced with fresh PBS. The cumulative RA release was estimated using the following equation [36]:(3)$$\mathrm{Cumulative}\;\mathrm{percentage}\;\mathrm{RA}\;\mathrm{release}\;\left(\%\right)=\frac{\mathrm{volume}\;\mathrm{of}\;\mathrm{sample}\;\mathrm{withdrawn}}{\mathrm{volume}\;\mathrm{of}\;\mathrm{release}\;\mathrm{medium}}\;\times \;P_{t-1}+P_{t}$$
where is $P_{t-1}$ is the percentage release at *t* − 1 time-point and $P_{t}$ is the percentage release at *t* time-point.

### 2.4. Biological Evaluations

#### 2.4.1. ROS-Glo H_2_O_2_ Assay

The ROS-Glo assay was performed to estimate the oxidative stress caused by the RA-loaded magnetite microspheres on the BHK—baby hamster kidney cell line. The samples were tested at the concentration of 0.05 mg/µL, at two time-points, i.e., 24 and 72 h. The H_2_O_2_ levels were measured using the ROS-Glo assay (Promega), according to the instructions of the manufacturer. Bioluminescent measurements were performed using a SpectraMax i3x Multi-Mode microplate reader and the SoftMax Pro 6 software and the results were expressed as record relative luminescence units (RLU).

#### 2.4.2. MTT Assay

The MTT assay was performed on a HepG2—human hepatocarcinoma cell line. Prior to the testing, all samples were subjected to a 1 h sterilization using UV light. The samples were tested at the concentration of 0.05 mg/µL, at two time-points, i.e., 24 and 72 h. After removing the culture medium, the cells were washed with PBS and dark-incubated for 2 h with (3-(4,5-dimethylthiazol-2-yl)-2,5-diphenyltetrazolium bromide solution at 37 °C and 5% CO_2_. Furthermore, the MTT solution was removed, and the formazan crystals were resuspended in isopropanol until complete solubilization. The absorbance was spectrophotometrically determined at 595 nm using a SpectraMax i3x Multi-Mode microplate reader and the SoftMax Pro 6 software. The cell viability of the samples was calculated using the following equation:(4)$$\mathrm{Cell}\;\mathrm{viability}\;\left(\%\right)=\frac{\mathrm{OD}595\;\mathrm{sample}}{\mathrm{OD}595\;\mathrm{control}}\;\times \;100$$

#### 2.4.3. Statistical Analysis

All biological experiments were performed in duplicate; thus, data are represented as the mean ± standard deviation. The statistical analysis was performed using the GraphPad Prism 9 software (San Diego, CA, USA), and the comparisons were made using the one-way analysis of variance (ANOVA) and, subsequently, the two-tails *t*-test with Bonferroni post hoc correction (*p* < 0.05).

## 3. Results

The present study aimed to investigate the efficiency of RA-loaded magnetite microspheres in anticancer applications. The magnetite microspheres were obtained through the solvothermal method, using PEG with two different molecular weights (Mw = 8000 g/mol and Mw = 20,000 g/mol). Therefore, two types of magnetite microspheres were obtained, which were further used as carriers for RA loading.

In this context, the first characterization method employed was XRD in order to demonstrate the formation of magnetite as the mineralogical phase (Figure 2). Subsequently, the Rietveld refinement allowed for assessing the average crystallite size and the unit cell parameters of the two types of the obtained microspheres (Table 2). As can be seen, the diffractograms demonstrate the formation of magnetite in the Fd-3m cubic crystal system as the unique mineralogical phase through the characteristic Miller indices (JCPDS 04-008-4511 [37]). Furthermore, the considerably high intensity values for each diffraction peak prove a high degree of crystallinity of the microspheres, as compared to nanoscaled magnetite [38,39,40]. The Rietveld refinement results show the same dimensions of the unit cells, but a larger crystallite size for the 20000 sample.

The morphology of the two samples was assessed through SEM (Figure 3). As the micrographs show, both samples exhibit a uniform spherical shape with dimensions in the submicronic range. Furthermore, there are no significant differences between the two types of microspheres. For a more precise analysis of the microsphere dimensions, the acquired micrographs were used for assessing their size distribution by measuring 350 individual spheres in the ImageJ software. Subsequently, the acquired measurements were used for creating the size distributions of each sample (Figure 4). It can be observed that the size distribution for the 20000 sample is narrower, ranging between 150 and 300 nm, in contrast with the 8000 sample, which ranges between 125 and 350 nm. Additionally, the Lorentz fitting curve demonstrated similar mean dimensions for the two types of microspheres, namely 252.85 ± 3.41 nm and 260.84 ± 3.98 nm for the 8000 and 20000 sample, respectively. These values are in accordance with the average crystallite sizes, proving that the obtained magnetite is monocrystalline.

The FT-IR analysis was employed in order to establish the bonds and functional groups present within the so-obtained samples. In this context, Figure 5 depicts the FT-IR spectra acquired for both pristine and RA-loaded magnetite microspheres. The formation of magnetite was demonstrated anew through the presence of the Fe-O bond at 525–535 cm^−1^. The addition of RA within the systems leads to the shift of this bond to the right and a decrease of the absorption peak intensity, which are generally associated with physical adsorption phenomena at the surface of the carriers. On one hand, the absorption bands from 650 to 3000 cm^−1^ within the pristine samples were attributed to PEG, which is characterized by the presence of aliphatic ether groups (C-O stretching between 1000 and 1200 cm^−1^), C-H bending between 1250 and 1500 cm^−1^, and C-H stretching between 2750 and 3000 cm^−1^ [41]. On the other hand, the presence of RA is demonstrated through the absorption bands between 550 and 1000 cm^−1^ characteristic to the aliphatic configurations of the C-H bond that are part of the compound fingerprint.

The stability of the pristine and RA-loaded magnetite microspheres was assessed through the hydrodynamic diameter and zeta potential measurements (Figure 6). Except for the 8000_1% sample, the addition of RA within the drug delivery systems leads to an increase of the hydrodynamic diameter due to a higher agglomeration tendency and an increased interaction between the hydroxyl groups present within the structure of RA and water molecules. Moreover, it can be seen that the highest polydispersity index values were registered for the pristine magnetite microspheres, thus proving their polydisperse nature. Nonetheless, increasing the RA concentration to 5% significantly reduces the polydispersity index to 0, followed by a subsequent increase for the 10% concentrations. These observations are in correlation with the zeta potential values, which are close to 0 mV for all samples and, thus, confirming the agglomeration tendency.

The final physicochemical characterization employed was based on the assessment of the LC%, EE%, and the drug release studies through UV–Vis spectrophotometry (Table 3 and Figure 7). As expected, the LC% increases with the amount of RA added due to a higher probability of the RA molecules interacting with the magnetite microspheres. Moreover, since the EE% decrease is not proportional to the amount of RA, it can be said that the surface of the magnetite microspheres is not yet saturated. Thus, EE% could be increased by prolonging the interaction time between RA and the microspheres. Furthermore, results show that the use of a lower molecular weight PEG leads to a higher capacity to adsorb RA onto the surface of the microspheres, despite there not being significant differences in terms of size. However, the amount released is larger for the higher molecular weight PEG samples, which could mean that it forms weaker bonds with RA.

The release study was conducted for 21 days in order to demonstrate the sustained release of RA from the magnetite-based drug delivery systems (Figure 6). As the release curves show, all samples reached a sustained release at the 7-day/14-day time intervals. While the cumulative release is proportional to the amount of RA loaded in both cases, there are several differences that are considerably important. First, as previously observed, the release from the 8000 samples is considerably lower, probably due to stronger bonds between the bioactive compound and the magnetite carriers. Second, the difference between the 20000_1% and 20000_5% samples is minor, both achieving a ~50% cumulative release. Third, minor fluctuations in the release curves can be observed for almost all samples, and they could be attributed to a non-homogenous diffusion of RA within the release media.

Finally, the biological evaluation of the RA-loaded magnetite microspheres involved assessing both their oxidative stress-related behavior upon non-cancerous cells as a biocompatibility indicator and their anticancer potential.

The oxidative stress was evaluated through the ROS-Glo assay on the BHK cell line (Figure 8). The 24 h time point shows higher values associated with an increased oxidative stress as compared to the control group. The presence of RA within the samples further enhances the oxidative stress levels. However, it is generally accepted that this time-point is not relevant for cytotoxicity assessments since the cells are stressed by significant changes in their environment. At the 72-h time-point, the oxidative stress levels increase for the RA-loaded systems, which is associated with the increased amount of RA released within the cell media, as shown by the UV–Vis measurements. Additionally, the oxidative stress is higher for the 20000 samples, which can also be correlated with the previously obtained results. In both cases, the lowest oxidative stress levels were registered for the 5% RA-loaded samples, which denote an optimal RA concentration. The possible mechanisms behind the generation of oxidative stress could involve two main pathways. On one hand, the magnetite microspheres could penetrate the cell membrane and subsequently generate reactive oxygen species through the Fenton and Haber–Weiss reactions [42,43]. On the other hand, studies have shown that the combined administration of RA and iron leads to the inactivation of the aconitase enzyme through the generation of hydrogen peroxide and superoxide anion. However, neither of the two components added alone did not cause or only caused limited amounts of reactive oxygen species [44]. Therefore, it could be assumed that the high levels of reactive oxygen species are a consequence of the synergistic effects of both RA and iron ion release from the magnetite microspheres. Additionally, the mechanisms involved could explain the variation in the results for the 72 h time points, as the reactive oxygen species production occurs upon the concomitant release of RA and iron ions. Nonetheless, the application of the present drug delivery systems within the organism should require a targeted release at the tumor site in order to prevent any side effects caused by the bioactive substance.

The anticancer potential of the RA-loaded magnetite microspheres was evaluated through the MTT assay on the HepG2 cell line (Figure 9). As the graph shows, the cell viability for the pristine magnetite microspheres is similar to the positive control, at both 24 and 72 h time-points, which represents another indicator of the magnetite carrier biocompatibility. The presence of the RA within the systems leads to significant increases after 24 h of exposure, which is associated to the initial burst release of the biocompound within the culture media in a quantity that promotes cell proliferation. By contrast, the cell viability at the 72-h time point drops considerably due to the increased amount of RA released which acts as an anticancer agent. In correlation with the previous UV–Vis and oxidative stress results, the cell viability for the 20000 samples is lower due to higher amounts of RA released. However, there are no significant differences between the concentrations of RA added. Thus, it can be concluded that the 24 h time point is not sufficient for achieving an efficient anticancer effect, since the small amounts of RA released are not cytotoxic, thus increasing the cell viability. This observation could also be correlated with the reduced levels of oxidative stress registered after 24 h. Moreover, as the results demonstrate, longer time periods allow for higher RA amounts to be released and to exert its anticancer activity.

## 4. Discussion

As the literature shows, the use of magnetite-based drug delivery systems for the treatment of a variety of diseases, especially cancer, is continuously growing [17]. Besides its well-known biocompatibility, magnetite is characterized by a high degree of versatility that allows for the control of its features by simply varying the synthesis method or the synthesis parameters [45]. However, the most commonly employed synthesis method is the co-precipitation of the iron ions into an alkaline medium. Despite its considerably simplicity, this synthesis method does not allow for the control of the particle size, shape, or surface reactivity [39,46,47,48]. Additionally, this method cannot usually be used for the development of particles with sizes in the (sub)micronic range [45]. There is a rising concern regarding the toxicity risks associated with nanoparticles since they strongly interact with the immune system, penetrate capillaries, and pose a threat due to the enhanced permeability and retention effect. In this context, researchers are focusing on increasing nanoparticle size by either forming micro-clusters consisting of thousands of nanoparticles or increasing the size of any individual particle [13].

Thus, this study aimed to develop submicronic magnetite microspheres which could overcome such concerns. The microspheres were synthesized through the solvothermal method, which involved the use of ethylene glycol as the solvent and, concomitantly, the reduction agent necessary for the transition from Fe^3+^ to Fe^2+^. Owing to its low toxicity, high abundancy, low cost, relatively high boiling point, and certain reducibility, ethylene glycol is a non-volatile and odorless green solvent widely used in chemical reactions [49,50], especially for the synthesis of monodisperse magnetite micro and nanoparticles [35,51,52,53,54,55]. Furthermore, the presence of NaAc within the reaction system plays two crucial roles, namely acting as an electrostatic stabilizer of the reaction and assisting the ethylene-glycol-mediated reduction of Fe^3+^. Previous articles reported the lack of particle formation as ethylene glycol alone could not reduce FeCl_3_ to Fe_3_O_4_ [35,51]. The presumed mechanism behind the formation of magnetite microspheres through the present experimental procedure involves the initial formation of ferrous and ferric hydroxide, followed by their reaction and transition to Fe_3_O_4_ [51].

Finally, PEG also plays a double role in the reaction mechanisms, both as the surfactant and for the prevention of particle agglomeration [35]. Nonetheless, PEG also acts as a coating agent to a certain degree, which could further prevent the issue associated with the oxidation of magnetite to maghemite [47,56,57]. In the context of the present study, the influence of the type of PEG in regard to its molecular weight on the morphology of the so-obtained microspheres was also investigated. Results proved that a higher molecular weight leads to a narrower particle size due to the 2.5-fold larger molecules. Moreover, the molecular weight of PEG directly influences the capacity to load and further release bioactive molecules. Specifically, results showed that a lower molecular weight PEG increases the LC% and EE% but decreases the cumulative drug release of the magnetite microspheres. This phenomenon could be explained by a stronger interaction and bonding between the 8000 g/mol PEG and RA.

Furthermore, the synthesis method presently employed also allows for the control of the magnetite carrier size. Precisely, as previous studies have suggested, the time and temperature of the reaction within the autoclave have a fundamental role in the outcome size of the spheres [35,51]. In this manner, the drug release profiles could be modified by the variation of the synthesis parameters.

Therefore, this study represents a step forward towards the possibility to elaborate synthesis procedures that can ensure a controlled release of drug molecules in anticancer therapies. By varying the type of the PEG surfactant and the time and temperature of the synthesis reaction, different drug release profiles could be achieved in accordance with the outcome application.

Finally, the choice of using RA within this study was based both on its well-known anticancer potential which could replace the synthetic chemotherapeutics currently used and on its beneficial effects for the organism, which could limit the side effects that are generally an important consequence of conventional anticancer treatments. To the best of our knowledge, there are no available studies investigating the release of RA from magnetite micro/nanoparticles. While studies based on RA-loaded drug delivery systems are considerably limited, the literature reports the utilization of polymeric nanoparticles, such as chitosan [58] and solid lipid nanoparticles [59]. Results demonstrate a lack of toxicological effects at low concentrations and decreased cell viability and risks of genotoxicity and necrosis in a concentration-dependent manner [59].

The future steps towards the development of this study would involve investigation of the magnetic field-mediated release of RA and, consequently, the hyperthermia potential of the drug delivery systems. Furthermore, the RA-loaded magnetite microspheres would be added to an injectable formulation and introduced directly into tumor models for the final in vivo study.

## 5. Conclusions

The present study aimed to develop efficient drug delivery systems based on magnetite microspheres for the controlled release of RA. Considering the issues associated with conventional cancer treatments, this study proposed an alternative that could limit side effects by using a natural anticancer agent and ensuring a controlled and sustained release. XRD results proved the formation of magnetite as the unique mineralogical phase, while SEM analysis showed the formation of particles with dimensions in the submicronic range. The successful loading of RA within the magnetite-based drug delivery systems was demonstrated through FT-IR, DLS and zeta potential, and UV–Vis spectrophotometry. Furthermore, the release profiles obtained through UV–Vis measurements showed the possibility to control the RA release by varying the type of surfactant. Biological evaluation showed limited generation of oxidative stress when applied to non-cancerous cells and promising anticancer potential against the HepG2 cell line. In this manner, this study represents a step forward in the development of drug delivery systems with the possibility to control drug release by varying the parameters of the solvothermal synthesis.

## Figures and Tables

**Figure 1 pharmaceutics-14-02292-f001:**
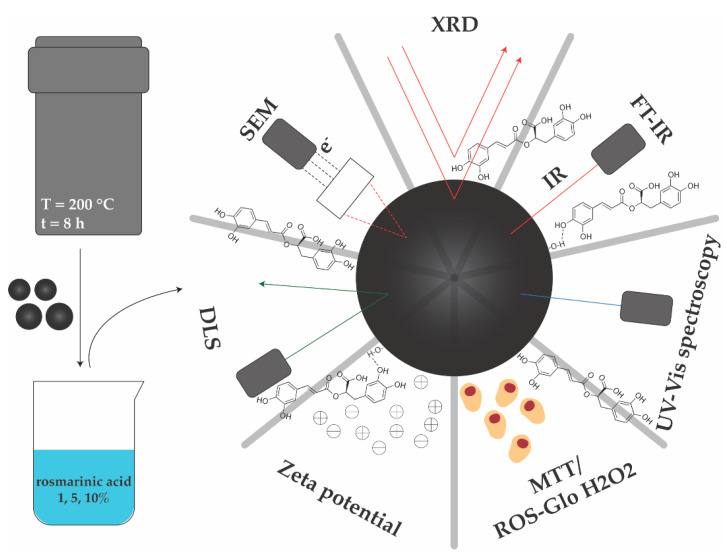
Schematic representation of the experimental design, involving the synthesis and characterization methods employed.

**Figure 2 pharmaceutics-14-02292-f002:**
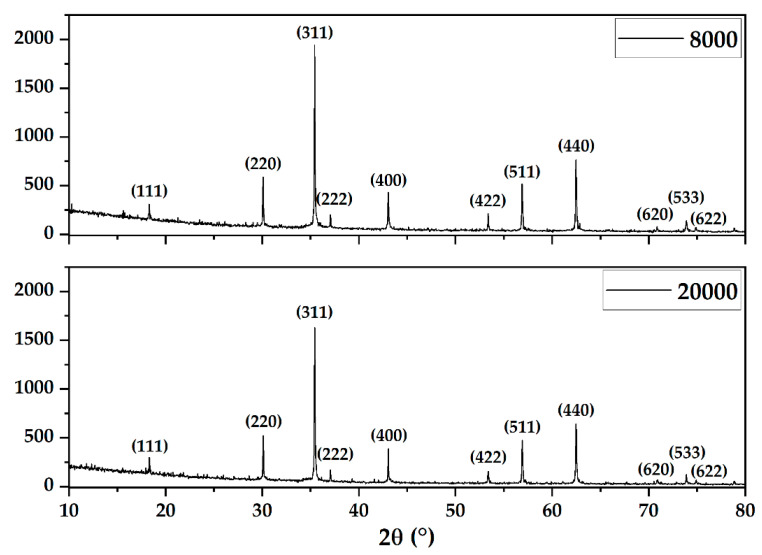
Diffractograms of the obtained magnetite microspheres.

**Figure 3 pharmaceutics-14-02292-f003:**
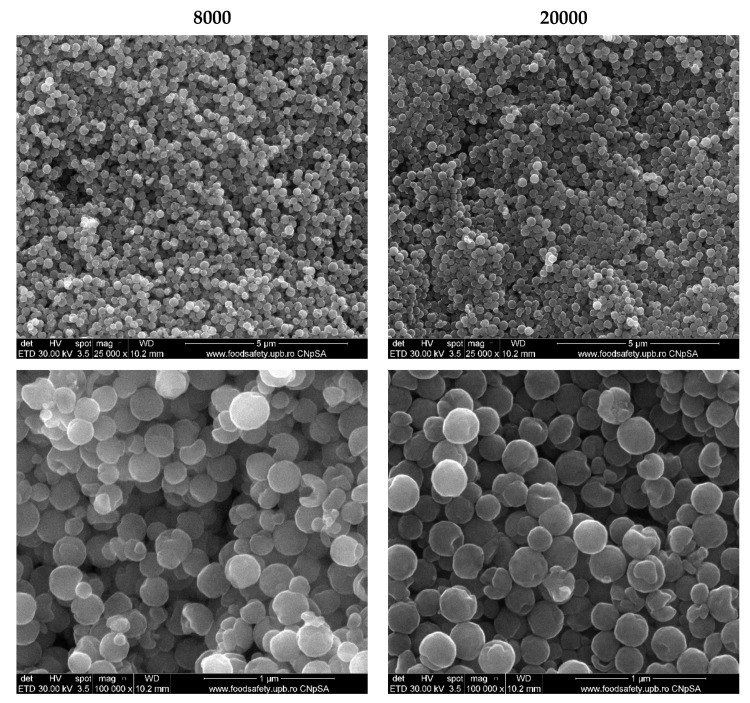
SEM images of the two types of magnetite microspheres.

**Figure 4 pharmaceutics-14-02292-f004:**
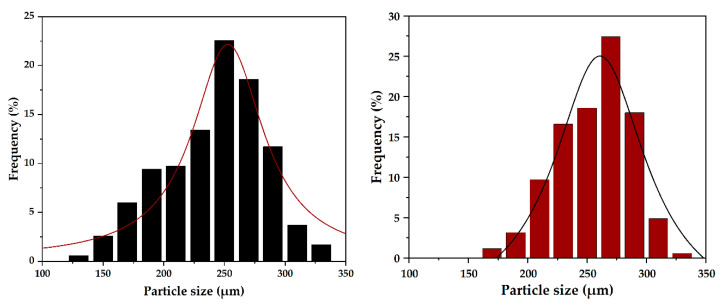
Size distributions and the fitting curves of the two types of magnetite microspheres (8000—**left** and 20000—**right**).

**Figure 5 pharmaceutics-14-02292-f005:**
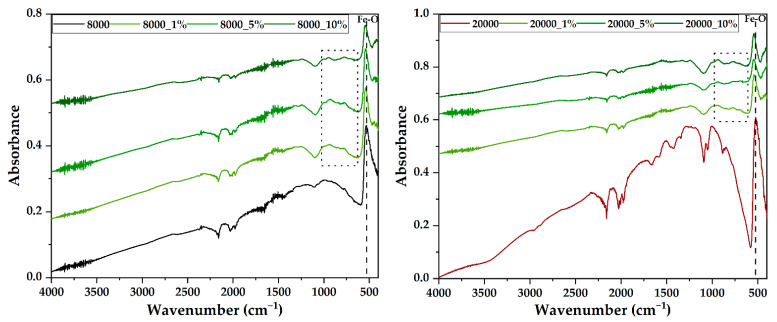
FT-IR spectra of the two types of magnetite microspheres (8000—**left** and 20000—**right**), both pristine and RA-loaded (marked square—the area of the RA fingerprint).

**Figure 6 pharmaceutics-14-02292-f006:**
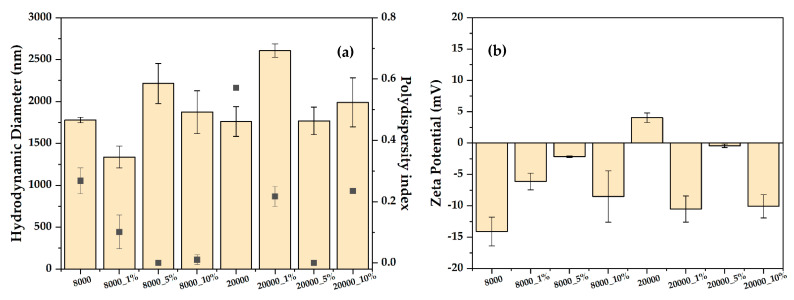
Hydrodynamic diameter (**a**) and zeta potential (**b**) values for the two types of magnetite microspheres, both pristine and RA-loaded (values are expressed as mean ± standard deviation).

**Figure 7 pharmaceutics-14-02292-f007:**
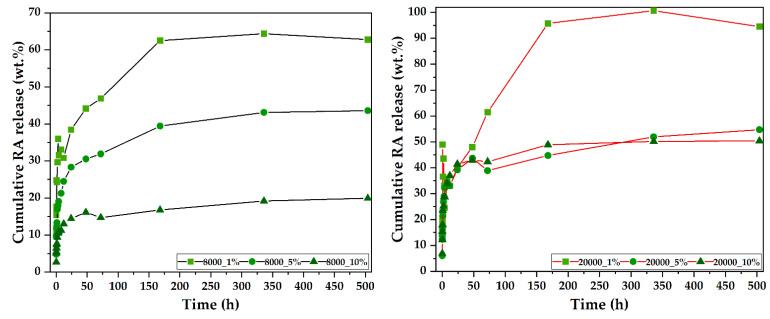
Cumulative release profiles at established time points for the RA-loaded magnetite microspheres (8000—**left** and 20000—**right**).

**Figure 8 pharmaceutics-14-02292-f008:**
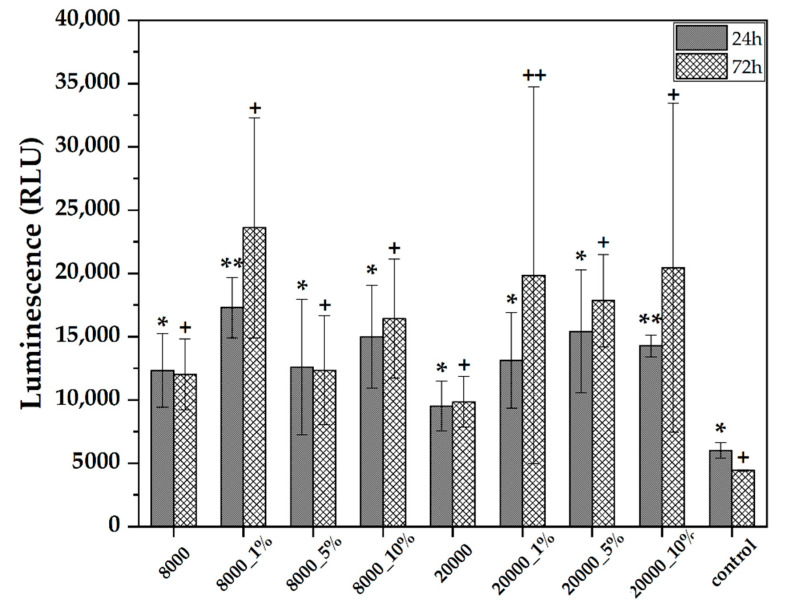
Oxidative stress results for the pristine and the RA-loaded magnetite microspheres (BHK cell line; values are expressed as mean ± standard deviation; different signs indicate significant differences between each sample, * and +—lower significance, ** and ++—higher significance).

**Figure 9 pharmaceutics-14-02292-f009:**
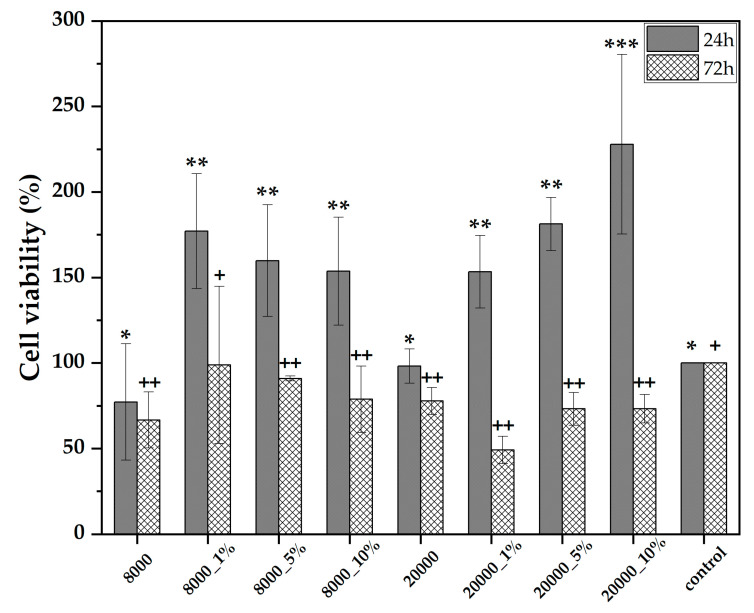
Cell viability results for the pristine and the RA-loaded magnetite microspheres (HepG2 cell line; values are expressed as mean ± standard deviation; different signs indicate significant differences between each sample, * and +—lower significance, **, *** and ++—higher significance).

**Table 1 pharmaceutics-14-02292-t001:** Summary of the samples obtained and their codes.

Type of PEG (g/mol)	RA Concentration (%)	Sample Code
8000	0	8000
	1	8000_1%
	5	8000_5%
	10	8000_10%
20000	0	20000
	1	20000_1%
	5	20000_5%
	10	20000_10%

**Table 2 pharmaceutics-14-02292-t002:** The cell parameters and the average crystallite sizes for the two types of magnetite microspheres.

Sample	Unit Cell Parameters						Average Crystallite Size [nm]
	a [Å]	b [Å]	c [Å]	α [°]	β [°]	γ [°]	
8000	8.40	8.40	8.40	90	90	90	218.44
20000	8.40	8.40	8.40	90	90	90	287.85

**Table 3 pharmaceutics-14-02292-t003:** The amount of RA added, the LC% and EE%, and the amount of RA released at specific time points for the drug delivery systems.

**Sample**	**Amount of RA Added (mg)**	**LC%**	**EE%**	**Amount of RA Released (mg)**		
				**24 h**	**72 h**	**21 Days**
8000_1%	4	0.50	50.34	0.19	0.24	0.32
8000_5%	20	0.93	18.50	0.26	0.30	0.41
8000_10%	40	1.83	18.27	0.27	0.27	0.37
20000_1%	4	0.31	31.36	0.13	0.19	0.30
20000_5%	20	0.78	15.55	0.31	0.30	0.43
20000_10%	40	1.16	11.59	0.48	0.49	0.59

## Data Availability

Not applicable.

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
