# Peer review of "Magnetite Microspheres for the Controlled Release of Rosmarinic Acid"

_pharmaceutics, 2022, doi:10.3390/pharmaceutics14112292_

Round 1

Reviewer 1 Report

Chircov et al. report the preparation of characterization of magnetite microspheres loaded with rosmarinic acid as potential anticancer agents. Overall, the manuscript is well-written and the research is sound, yet I have some comments. In some parts, the results are just presented but barely discussed or not discussed at all. I also have doubts about the novelty of this work. Thus, I would encourage the authors to improve the manuscript with some revisions. My comments are as follows:

1.     It would be nice to include an schematic that summarizes the essence of this paper and/or the design of the system.

2.     Intro: the relevance of rosmarinic acid and why it was chosen should be discussed.

3.     Intro: magnetite particles are not new, as the authors themselves acknowledge. Thus, the particular significance of this study and novelty aspects should be better discussed in the introduction.

4.     What is the mechanism behind RA incorporation on the particles? Is it adsorption? Entrapment? A clarification on this would be apreciated.

5.     Figure 2: The particles are below 1 um in size. Why are they called microspheres then? Why not to call them nanospheres through the manuscript?

6.     Figure 5: The DLS size is in the microscale, which may suggest particle inestability. Are this agglomerates stable? What does the error bar represent? Please indicate in the footnote. If possible, a plot with the PDI values should also be provided and discussed.

7.     What is the mechanism of the particles producing ROS? This could be better discussed.

8.     Figure 7: indicate in the footnote what the error bar represents.

9.     Figure 8: Why does RA induce an increase in cell viability?

10.  Discussion: Some paragraphs feel more like an intro than a discussion (e.g., RA properties) Please, move some of these to the intro and focus more on an integrated discussion of the specific results here.

11.  How does this work compare to others RA drug delivery systems? Please discuss previous works on RA delivery systems.

12.  Conclusions: The authors claim efficient anticancer potential? In any case, there is certain effect but just one treatment reaches viability below or near 50%. Please discuss appropriately.

Reviewer 2 Report

This is an interesting study and the authors have collected a unique dataset using various methodology and techniques. Overall, this is a clear, concise, and well-written manuscript. The introduction is relevant and theory based. Sufficient information about the previous study findings is presented for readers to follow the present study rationale and procedures.

1.       In Figure 5, for time point 72h, we see a massive standard division. How do you consider and discuss this issue?

2.        Based on the drug release profile, it seems 8000-1% and 20000-1% are the best. However, in MTT assay they look so toxic particularly at 72h timepoint. What is your strategy if you would examine the compound on In vitro / in vivo studies?

3.       In the ROS-Glo assay, both 8000-1% and 20000-1% have a considerable fluctuation after three days incubation which indicates the toxicity again. As far as the next step – eventually –  is to examine this drug carrier on cells, How this issue can be tackle?  

Round 2

Reviewer 1 Report

The authors addressed most of my comments.